# Advances in Trace Element Supplementation for Parenteral Nutrition

**DOI:** 10.3390/nu14091770

**Published:** 2022-04-23

**Authors:** Patti Perks, Emily Huynh, Karolina Kaluza, Joseph I. Boullata

**Affiliations:** 1Nutrition Services, University of Virginia Children’s Hospital, Charlottesville, VA 22903, USA; 2American Regent, Inc., Norristown, PA 19403, USA; ehuynh@americanregent.com (E.H.); kkaluza@americanregent.com (K.K.); 3College of Pharmacy and Health Sciences, Queens Campus, St. John’s University, Jamaica, NY 11439, USA; 4Previously with the Clinical Nutrition Support Services, Hospital of the University of Pennsylvania, Philadelphia, PA 19104, USA; jb4consult@comcast.net

**Keywords:** adults, children, copper, manganese, micronutrient, parenteral nutrition, selenium, trace elements, zinc

## Abstract

Parenteral nutrition (PN) provides support for patients lacking sufficient intestinal absorption of nutrients. Historically, the need for trace element (TE) supplementation was poorly appreciated, and multi-TE products were not initially subjected to rigorous oversight by the United States Food and Drug Administration (FDA). Subsequently, the American Society for Parenteral and Enteral Nutrition (ASPEN) issued dosage recommendations for PN, which are updated periodically. The FDA has implemented review and approval processes to ensure access to safer and more effective TE products. The development of a multi-TE product meeting ASPEN recommendations and FDA requirements is the result of a partnership between the FDA, industry, and clinicians with expertise in PN. This article examines the rationale for the development of TRALEMENT^®^ (Trace Elements Injection 4*) and the FDA’s rigorous requirements leading to its review and approval. This combination product contains copper, manganese, selenium, and zinc and is indicated for use in adults and pediatric patients weighing ≥10 kg. Comprehensive management of PN therapy requires consideration of many factors when prescribing, reviewing, preparing, and administering PN, as well as monitoring the nutritional status of patients receiving PN. Understanding patients’ TE requirements and incorporating them into PN is an important part of contemporary PN therapy.

## 1. Introduction

Parenteral nutrition (PN) therapy is an essential intervention for patients with insufficient intestinal absorption of nutrients that are provided orally or enterally [1]. PN delivers an admixture of amino acids, carbohydrates, lipids, vitamins, electrolytes, and trace elements (TEs) directly into the bloodstream [2]. Individuals of all ages who are unable to absorb enterally administered nutrients can benefit from PN to provide nutrition support for short- and long-term periods of time [1]. Each year in the United States (US), approximately 300,000 patients receive PN during hospitalization [3], and approximately 25,000 additional patients receive PN at home [4].

The prescribing and preparation of PN admixtures require careful patient assessment to ensure that a patient’s individual nutritional requirements, including for TEs, are met [5,6]. TEs are required in very low concentrations, but inadequate or excess intake can result in clinical complications, consequently, under- and overdosing has potentially serious consequences [7].

PN was introduced and developed as a medical therapy in the 1960s and 1970s [8]. Micronutrient deficiency syndromes were described in patients receiving long-term PN, and the need for multivitamin and multi-TE products was recognized during this period. Initially, the Nutrition Advisory Group of the American Medical Association (NAG-AMA) made recommendations regarding micronutrient requirements to the US Food and Drug Administration (FDA). Subsequently, the American Society for Parenteral and Enteral Nutrition (ASPEN) assumed the lead role in this regard and continues to provide up-to-date recommendations for dosing of PN components including TEs. 

Commercial, micronutrient-containing products were not available during the initial development of PN [8]. At first, hospital pharmacies compounded single-entity TE-containing solutions, then NAG-AMA designed TE recommendations to be used to formulate a product for use in PN [8]. Over time, reports of TE toxicity in patients receiving PN prompted ASPEN to issue updated recommendations for TE supplementation and to call for safer parenteral micronutrient products to be developed. 

Applying uniform standards to multi-TE products is an ongoing challenge. Multi-TE products used in the preparation of PN are considered drugs by the FDA; however, due to the unique manner in which PN has developed, commercial multi-TE products were historically considered to be “marketed unapproved drugs.” Implementing recent ASPEN recommendations is a challenge, because changes to commercial formulations now require product submissions and approval by the FDA. This may be perceived as a disincentive by manufacturers who could discontinue a product rather than incurring the expense of product modification. The situation is compounded by the frequent occurrence of TE shortages, which have necessitated the FDA to permit temporary authorization of products from Europe, which do not contain the type and amount of TEs recommended by ASPEN. 

Against this backdrop, the development of a multi-TE product that meets ASPEN recommendations and FDA regulatory requirements is the result of an important partnership between the FDA, industry, and clinicians with expertise in PN. The aims of this article are to describe the rationale for the development of TRALEMENT^®^ (Trace Elements Injection 4*, hereafter referred to as Tralement; American Regent, Inc., Norristown, PA, USA), current ASPEN recommendations for TE dosing in PN, FDA regulatory requirements, and the New Drug Application (NDA) process that was pursued in the approval of Tralement.

## 2. TEs and Their Physiological Roles

Micronutrients are as important to patient care as macronutrients. Besides vitamins, the minerals include TEs, which are inorganic micronutrients present in the diet in very small quantities. In general, TEs play important and diverse physiologic roles [9], which involve metalloenzyme cofactors, intracellular signaling, gene expression, and free radical scavenging (Table 1) [10,11,12]. Of the dozen TEs important for biological function, recommended dietary allowances (RDAs) or adequate intake (AI) levels have been established for oral intake of these 9: chromium, copper, fluorine, iodine, iron, manganese, molybdenum, selenium, and zinc [13]. For a detailed review of TEs and their role and properties in medical nutrition therapy, please refer to the updated ESPEN micronutrient guidelines [14].

Five of these TEs (chromium, copper, manganese, selenium, and zinc) are routinely included in PN, as per US recommendations [12,13]. Intravenous (IV) administration of suboptimal or excessive amounts of these TEs can disrupt a range of physiologic processes, causing deleterious clinical signs and symptoms, which are summarized briefly in Table 1 [10,11,12]. TEs are incorporated into PN admixtures, either individually, in a fixed-dose TE combination product, or both when indicated [7].

## 3. TE PN Formulations in the US

### 3.1. A Historical Perspective

Many parenteral TE products, available since the 1970s and 1980s in the US, were in use before an FDA regulatory approval was required and are designated as “marketed unapproved drugs” (Table 2) [15,16,17,18,19,20,21,22,23,24]. As such, there was limited FDA oversight of their manufacturing, little or no safety and efficacy data available, and no prescribing information to guide prescribers in the use of these products [25]. For over 3 decades, “marketed unapproved” fixed-dose multi-TE products were the sole source of TE components for patients receiving PN [25,26]. Although these products contained a combination of either 4 or 5 TEs (chromium, copper, manganese, with or without selenium, and zinc; Table 2), they became outdated as clinical recommendations evolved [25]. For example, Multitrace-5 (American Regent, Inc.) was a “marketed unapproved” fixed-dose multi-TE product that contained all 5 of the aforementioned TEs [18], but the concentrations of copper and manganese were excessive. The FDA could have removed Multitrace-5 and other marketed unapproved multi-TE products from the market before approved alternatives became available; however, removing such products would have reduced the supply of available TE products and may have precipitated or exacerbated a shortage [25]. Instead, a smaller volume of the product was used to limit doses of copper and manganese. Mutitrace-5 has since been discontinued. 

Some “marketed unapproved drugs”, including PN components, were grandfathered as a result of existing FDA regulations; consequently, these products continue to be manufactured with limited FDA oversight and marketed without prescribing information to guide dosing and inform prescribers about important product safety-related concerns [25]. This lack of FDA oversight has been a source of concern in the PN community, particularly with regard to the potential for toxicity associated with higher-than-recommended doses of some TEs in these formulations [8]. Additionally, the FDA does not oversee the labeling of contaminant elements in PN component products with the exception of aluminum [27].

Concerns about the micronutrient content of PN preparations were raised with the FDA as early as 1972 (Figure 1) [8]. Guidelines on the TE content of PN for adults and children were first developed by the NAG-AMA in 1979, recommending the inclusion of chromium, copper, manganese, and zinc [28]. These guidelines were updated in 1984 with changes to the recommended doses of chromium and manganese [29,30] and a recommendation that patients on long-term or home PN receive selenium supplementation [31]. Additional pediatric guidelines by the American Society for Clinical Nutrition were published in 1988 (Figure 1) [32]. Dosages of various TEs (principally chromium, manganese, and selenium) were revised in subsequent guidelines [2,33,34], leading ASPEN to form a Novel Nutrient Task Force in 2009, with the aim of developing an evidence-based position statement on micronutrient requirements in PN [13]. This position statement was published in 2012 [8] and reemphasized by the Task Force in a “Call to Action” in 2015 for safer PN component products in the US, based on reports of TE toxicity in patients receiving PN [8].

### 3.2. Current Recommendations for TE PN Formulations

Concise adult dosing recommendations for TEs in PN were released by ASPEN in 2019 based on the 2012 position statement [35]; the dosages of TEs outlined in these recommendations are summarized in Table 3 [35,36]. The ASPEN recommendations for pediatric patients include guidance for adolescents, children and infants, and preterm neonates. The focus of this review is the formulation and approval process of a multi-TE product developed for use in adults, and adolescents and children/infants weighing ≥10 kg. ASPEN provides dosage recommendations based on age and body weight for infants and children (10–40 kg) and adolescents (>40 kg) (Table 3) [35,36]. Following the development of Tralement, the FDA also approved MULTRYS ™ (American Regent, Inc.), indicated for neonatal and pediatric patients (<10 kg) with special needs as a source of copper, manganese, selenium, and zinc in PN [24].

## 4. Advantages of Fixed-Dose TE Combination versus Single-Entity TE PN Products

TEs can be incorporated into PN from fixed-dose TE combination products, single-entity TE products, or both (Table 4) [37]. Advantages of the fixed-dose combination products include: (1) a simplified preparation process; (2) a reduced risk of errors with dosage calculations; and (3) a lower risk of contamination given fewer steps to prepare PN for an individual patient. More detailed explanations of risks associated with TE combination products can be found in the current ESPEN micronutrient guidelines [14]

The single-entity TE PN formulations provide more flexibility with dosing for patients with varying TE needs (Table 4) [37]. Patients may require different amounts of individual TEs because of underlying medical conditions and/or low micronutrient status, thus necessitating TE supplementation in addition to those supplied from a fixed-dose TE combination product (Table 5) [37]. Moreover, single-entity TE supplementation may be indicated in those losing abnormally high amounts of micronutrients as a consequence of burns, vomiting/diarrhea, fistula or ostomy output, wound drainage, or when it becomes necessary to exclude one TE and provide the others [13].

To adequately meet a patient’s TE needs, a fixed-dose TE combination product can provide basal TE doses, while single-entity TE products can be used to individualize the dose of a particular TE or TEs when clinically indicated. For example, if the required dose of one particular TE is higher than that provided in a fixed-dose TE combination product, the dose can be achieved by adding the required amount of a single-entity TE product to the PN admixture in addition to the fixed-dose TE combination product. Such an approach highlights the importance of having a safe and effective fixed-dose TE combination PN formulation that not only adheres to established TE recommended dosing guidelines but also meets the TE needs of a majority of patients.

Alternatively, some patients may occasionally need less of an amount of a particular TE than the levels contained in a fixed-dose TE combination product, for example, copper or manganese. In such situations, the fixed-dose TE combination product cannot be used, and the individual TE requirements must be provided by using single-entity TE products alone.

To ensure patients have access to effective and safe TEs, it is critical that both fixed-dose TE combination products as well as single-entity TE products undergo a thorough FDA regulatory review and approval process. In 2020, following a rigorous, multistep regulatory review and approval process, the FDA granted regulatory approval of Tralement [37]. This product was developed in compliance with established parenteral TE recommendations with the intent of meeting the TE needs of a majority of patients while aiming for a conservative approach to avoid excessive TE dosing. This approach allows for addition of single-entity TE products to individualize to a patient’s needs.

## 5. Tralement: Rationale and FDA Regulatory Review and Approval

As early as around 2009, ASPEN advised the FDA of their concern about the inadequate regulatory review and approval process for PN component products and the continued marketing of unapproved TE products [8,37]. ASPEN also stressed that the approval of any safe and effective TE product necessitates compliance with the clinical practice standards as well as adherence to FDA quality standards.

The FDA regulates PN component products as injectable drugs and applies stringent regulatory requirements [25]. Such requirements are necessary because of the products’ IV route of administration and use in individuals with serious health conditions [25]. Considerable evidence of the safety and efficacy of PN component products is required, usually from controlled clinical trials; however, the FDA may consider evidentiary clinical safety/efficacy data reported in the scientific literature [25]. The FDA has two alternative NDA pathways for new drug approvals, including for PN component products [24]. One pathway requires safety and efficacy data derived from clinical trials conducted by the drug sponsor (505(b)(1) NDA). The other pathway allows the sponsor to cite existing safety and efficacy literature and/or clinical trial data previously reported to the FDA by another drug sponsor (505(b)(2) NDA).

The latter pathway (505(b)(2) NDA) was used for Tralement [37,39]. The NDA cited published literature on copper and manganese, included data from previously approved NDAs for selenium [40] and zinc [41], and was consistent with ASPEN recommendations for adults and pediatric patients weighing ≥10 kg [23,37].

### 5.1. Literature Assessment

The literature assessment identified three key components including: a rationale for including each individual TE in a standard PN formulation; current ASPEN dosing recommendations for each TE in adult and pediatric patients; and evidence to support the safety and efficacy of the proposed dosing for each individual TE [37].

Collectively, the review included a summary of data pertaining to the daily requirements for TEs in patients receiving PN, enteral intake of TEs, and available TE balance studies; the recommended daily dosages of the individual TEs; and blood concentration and tissue level data for each individual TE. Information pertaining to TE deficiencies, dietary standards such as RDA [42], and dietary reference intakes [43] were also considered (Figure 2) [37].

The proposed dosing for Tralement stemmed from a synthesis of data included in 18 systematic literature reviews (SLRs), 10 of which assessed parenteral TE exposure-response relationships, and 174 published studies on TE exposure in adult (*n* = 125) and pediatric cohorts (*n* = 49) reported.

The product’s indication in adult and pediatric patients as a source of each respective TE for PN was granted approval on the basis of: (1) clinical data on TE supplementation in PN; (2) the enteral requirements (e.g., RDA) of the respective TEs including relative bioavailability; (3) current clinical guidelines for PN; and (4) each element’s toxicity profile, as well as the time and extent of the respective TE’s use in clinical practice.

### 5.2. Studies on Product Sterility, Compatibility, and Stability in PN Admixtures

Admixture studies using Tralement were conducted under aseptic conditions to evaluate its stability and compatibility with common commercially available PN admixtures (amino acids in dextrose, electrolytes, and vitamins, with and without lipid emulsions). In the 4 admixture studies, Tralement (1 mL) was added to 2-L IV infusion bags of Kabiven^®^ and Clinimix^®^ E 4.25/5 sulfite-free injections; the bags were then refrigerated (2–8 °C) for 14 days. The studies tested for pH change (i.e., compatibility), chemical retention of each TE (i.e., stability), and for any adventitious microbial contamination growth (i.e., sterility) during the preparation and storage of the PN admixtures with Tralement. The results of these studies met the protocols’ acceptance criteria and support the US Pharmacopeia General Chapter <797> medium risk storage [44].

## 6. Practical Considerations for Clinical TE PN Management

As previously discussed, a fixed-dose multi-TE combination product that complies with current parenteral TE recommendations for copper, manganese, selenium, and zinc, has the benefit of addressing the TE nutritional needs of a majority of patients. Pediatric patients require TEs to support growth and development in addition to replacing losses attributed to surgical and/or medical conditions. TEs should be included in the initial prescription for PN [36]. Additional single-entity TE products can be included in the PN formulation to individualize TE dosing. For example, infants and children with congenital cardiac disease may have increased TE requirements because of chylous effusions and drainage or chronic diuretic therapy [36,45].

Considering that PN is a complex therapy, it is important to weigh the advantages and disadvantages of single-entity TE versus fixed-dose multi-TE combination products and the potential presence of contaminants in other PN components. Established standardized processes for preparing PN formulations to address individual patient TE needs as well as clinical practices to monitor TE status are critical for tailoring PN to maintain an individual patient’s nutritional status and overall health.

### 6.1. PN Admixture Preparation

The process for compounding PN admixtures adheres to a prescription customized for an individual patient’s requirements or based on a standardized formulation [5,46]. Whether the PN admixtures are prepared manually or by an automated compounding device, accurate PN preparation and labeling, including the beyond-use date, and appropriate storage conditions are essential [46]. Because TE contaminants from other PN components can affect an admixture’s total TE content, awareness of all potential sources of contamination is important [5].

PN admixtures may contain TEs that originate from sources other than intentionally added TE products and may be considered contaminants. Sources of TE contaminants may include amino acids, calcium salts (i.e., calcium gluconate and chloride), and electrolytes (e.g., potassium and sodium phosphate), as well as TE formulations [47,48,49]. TE contaminants can increase the total amount of a particular TE that a patient is receiving and result in exposure to excessive amounts over time. Clinically significant amounts of manganese have been detected in PN admixtures for neonates and adults, and accumulation of manganese in patients receiving PN is associated with neurotoxicity [50]. The amount of manganese in PN prepared using individual TE ingredients not including Mn was approximately 10% of that contained in PN prepared with a standard fixed-dose neonatal multi-TE product that contained manganese [51]. The authors of this study did not identify the source of manganese contamination [51]. A study in the United Kingdom found elevated manganese levels in 30% of adult patients who were receiving manganese-free PN [52].

Notably, for fixed-dosed TE combination PN formulations, limits have been established on the number of components in the admixture to minimize TE contamination in the final PN infusate [13]. For example, the TE chromium may be present as a contaminant, possibly introduced from other PN components [8,47]. During product selection and development of Tralement, the literature was evaluated and PN solutions were assessed for chromium content. It was concluded that the contaminant level of chromium in most parenteral solutions would meet the estimated requirement for chromium, and therefore, the FDA-recommended formulation did not warrant the inclusion of chromium [23,37].

In addition to determining the appropriate dose of each component in PN, the compatibility and stability of admixed components and the stability of the final PN admixture are important considerations [34,53]. TEs are included in the PN prescription based on the patient’s clinical needs and the available compatibility and stability data. TEs are generally stable at the recommended doses in common PN admixtures. However, it is important to appreciate that although analytical methods can determine the final mineral concentration, they do not necessarily distinguish between oxidation states or complexation and precipitation [53,54]. PN admixtures are rich in nutrients and, therefore, an ideal media for microbial growth if contaminated; however, standard practice for preparation mandates aseptic conditions, reducing the risk of microbial contamination [34,53].

### 6.2. PN and TE Shortages in the US

In the past, serious and persistent shortages resulted in the prescription of less than adequate doses of certain components in PN [35]. This situation prompted ASPEN to include the following extraordinary statement in its 2019 dosing recommendations: “Clinicians who have entered practice within the last 10 years may have never cared for patients receiving PN therapy without a shortage of PN components.” [35]. In addition, ASPEN advised clinicians to resort to rationing only during shortages, and to return to appropriate dosing as soon as the component shortage had been resolved [35]. When TE components were in short supply, the FDA approved select European multi-TE products for temporary importation to alleviate the shortage [8]. However, neither the adult (Addamel N^®^) nor the pediatric product (Peditrace^®^) imported from Europe provided the identity or quantity of TEs specified in the ASPEN recommendations [8]. Moreover, these products contained additional components that were not present in US formulations (the adult product contained iron, molybdenum, iodide, and fluoride, whereas the pediatric product contained iodide and fluoride) [8]. During national shortages of individual TEs, some centers instituted changes to PN practices that were not well supported. Such practices have included prioritizing supplies for pediatric patients, reducing doses or omitting some TEs from PN admixtures, or diluting fixed-dose TE combination PN formulations for pediatric use. With such practices in place, some patients developed severe biochemical deficiencies [55]. These shortages and the attempts to cope with them highlight the need for a safe and reliable source of TE components that comply with US recommendations and regulatory standards.

### 6.3. Conditions Affecting TE Needs

#### 6.3.1. Increased TE Requirements

As a general rule, the greater the severity of a patient’s clinical condition, the greater the potential for micronutrient imbalances, thus affecting TE needs (Table 5) [12,36,38]. Burns in particular increase TE requirements (especially for copper, selenium, and zinc) as a result of exudative losses and the associated oxidative stress and inflammation. TE supplementation is considered essential for aiding wound healing and reducing the risk of infection [38,56]. Moreover, many conditions or interventions, such as enterocutaneous fistulas, externalized drains, ostomy output, diarrhea, diuretics, and renal replacement therapy, also result in increased TE requirements [38]. Infants and children with intestinal failure or ostomies after partial bowel resection require higher-than-recommended doses of copper (>20 mcg/kg/day) and zinc (500 mcg/kg/day) in PN as a result of intestinal losses or malabsorption [57,58]. Selenium needs may also be increased after intestinal surgery [36]. A fixed-dose multi-TE product can be used as a base to which single-entity TE products are added so that a given patient’s global TE requirements can be met.

#### 6.3.2. Decreased TE Requirements

Cholestasis or other forms of hepatobiliary dysfunction can lead to the accumulation of copper and manganese, so patients with these conditions warrant monitoring and may need a dose reduction in one or both of these TEs (Table 5) [7,12,13]. Long-term use of PN has been associated with the development of hepatic dysfunction, cholestasis, and potentially gallbladder disease and steatosis [59]. Approximately 60% of hospitalized patients receiving PN for at least 2 weeks have been shown to develop these types of hepatic disorders [60]. However, discontinuing copper in patients with cholestasis may lead to copper deficiency, so it is advisable to adjust the PN copper dose based on serum concentrations and the clinical scenario [36,61,62]. More frequent monitoring of TE status is required with long-term PN whether in the chronic critical care setting or at home, but there are no established guidelines for the frequency of monitoring in either setting [1,63]. Monitoring of the TE status once every 6 months in the absence of symptomatic presentation would be reasonable for adults. A multi-TE product is not suitable for patients who require quantities of one or more components that are lower than the amount contained in the product. The TE requirements of these patients must be provided by using single entity TE products to ensure appropriate dosing of all TEs.

### 6.4. Patient Monitoring

The risk of suboptimal or supraphysiologic TE levels due to prolonged PN is relatively low. However, reports of manganese toxicity and organ accumulation of manganese and copper in patients receiving long-term PN prompted ASPEN to reduce the recommended daily doses of these elements and serve as a reminder that patients must be closely monitored for signs and symptoms of TE deficiency or toxicity while receiving chronic PN [13]. Patients’ nutritional needs may change in response to changes in their clinical condition [13]. During long-term PN, subtle signs and symptoms of toxicity may be attributed to other conditions if TE accumulation is not considered [13]. If deficiency or toxicity is suspected for the following TEs, they can be monitored as follows [12]: copper—ceruloplasmin or serum copper levels; chromium—plasma or serum chromium levels (although they may not reflect tissue levels); manganese—whole blood manganese; selenium—serum selenium; and zinc—serum zinc levels. In addition to biochemical monitoring, clinical assessment of the patient remains paramount.

Considering the role that chromium plays in glucose metabolism, glucose intolerance may be indicative of chromium deficiency [12]. Glucose monitoring is important as PN is initiated and advanced; once dextrose delivery is at goal, monitoring should continue on a regular basis depending on the clinical indications for each patient. This glucose monitoring can provide sufficient monitoring for potential chromium deficiency, because serum chromium levels are a poor indicator of tissue levels [1,12]. The questionable role of parenteral chromium in glucose homeostasis is a subject for further research. Johnsen and colleagues recommend assessing the manganese status after 30 days in children receiving PN without TE supplementation, as many will require manganese supplementation at this time [64]. Manganese and chromium concentrations were frequently elevated in patients on long-term PN receiving previously available unapproved products [12]. Magnetic resonance imaging may be indicated in patients with suspected manganese toxicity to identify any deposition in the basal ganglia [12].

Inflammation can complicate the identification of a TE deficiency, because it may be associated with a redistribution of several micronutrients between the blood and tissue compartments and result in levels not necessarily reflective of the TE status [38]. Inflammation can lead to elevated levels of ceruloplasmin, which carries most of the circulating copper, and reduced levels of selenium and zinc, the latter of which is particularly affected by sepsis [12]. During an ongoing inflammatory state, the clinician’s assessment of TE status will need to combine nutrition history, physical examination, and biomarker measures unaffected by the acute inflammatory response [36]. Measuring C-reactive protein (CRP) levels along with plasma/serum levels of TEs may assist in interpretation of the TE status [38]. High levels of CRP may necessitate a reassessment of requirements and a need for increased doses of selenium in hospitalized patients receiving PN [65]. Ongoing evaluation of the patient’s status is strongly advised with appropriate modification of the PN prescription to match the patient’s needs [1].

## 7. Areas for Further Research and Education

The ASPEN position statement provides evidence-based guidelines on the dose of TEs for PN in stable adults and children. However, more clinical evidence is required to identify the TE requirements of patients with unique needs. Certain medical conditions can potentially affect a patient’s TE status; these include renal disease, liver disease, Wilson’s disease, pregnancy, short bowel syndrome or intestinal failure, gastrointestinal malabsorption, surgery, sepsis and inflammation, obesity, undernutrition, and advanced age. More clinical guidance is needed regarding the frequency of monitoring laboratory tests for specific TEs; the most efficacious and cost-effective testing; and problem solving for delivery of TEs in the home setting as well as during times of shortages. Further research is needed to determine whether additional TEs, such as iron, fluoride, and iodide, should be included in multi-TE products. The presence of TEs as contaminants in PN is an ongoing challenge. Further research is required to identify the sources of TE contamination, to determine whether these sources can be eliminated or better controlled, and how best to cope with this problem, especially in patients receiving long-term PN. Moreover, it is critical that efforts be made to improve educational training programs about PN prescription and coordination of PN therapy for healthcare providers. Additionally, order entry systems in electronic medical records need to be improved to provide consistency across healthcare settings.

## 8. Conclusions

There is an ongoing need for safe and effective PN component products to meet the complete nutritional requirements of patients receiving PN. ASPEN provides dosing recommendations for all PN components in order to ensure the safe and effective use of available PN components, including TE products. The FDA has a rigorous regulatory review and approval process for PN component products. Tralement received FDA approval via the 505(b)(2) NDA pathway. This product was developed in compliance with ASPEN recommendations and FDA regulations with the aim of providing the TE requirements of most adult and pediatric patients >10 kg receiving PN. Comprehensive management of PN therapy requires that healthcare professionals consider many factors when prescribing, reviewing, preparing, and administering PN, as well as monitoring the nutritional status of patients receiving PN. An understanding of patients’ TE requirements and how they are incorporated into PN is an important part of contemporary PN therapy. Consequently, healthcare professionals require comprehensive education and training to achieve competency in PN. Further research and education are needed to address existing gaps related to the delivery of TEs in PN, in particular for patients with unique TE requirements.

## Figures and Tables

**Figure 1 nutrients-14-01770-f001:**
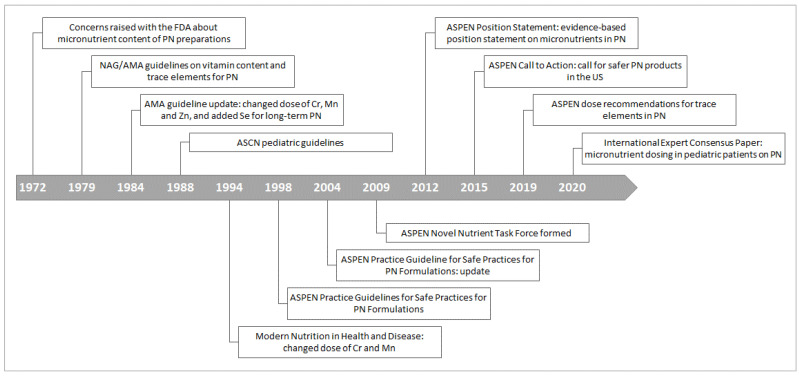
Timeline in the development of current US recommendations for dosing of trace elements in parenteral nutrition [8]. AMA, American Medical Association; ASCN, American Society for Clinical Nutrition; ASPEN, American Society for Parenteral and Enteral Nutrition; Cr, chromium; FDA, United States Food and Drug Administration; Mn, manganese; NAG, Nutrition Advisory Group; PN, parenteral nutrition; Se, selenium; US, United States; Zn, zinc.

**Figure 2 nutrients-14-01770-f002:**
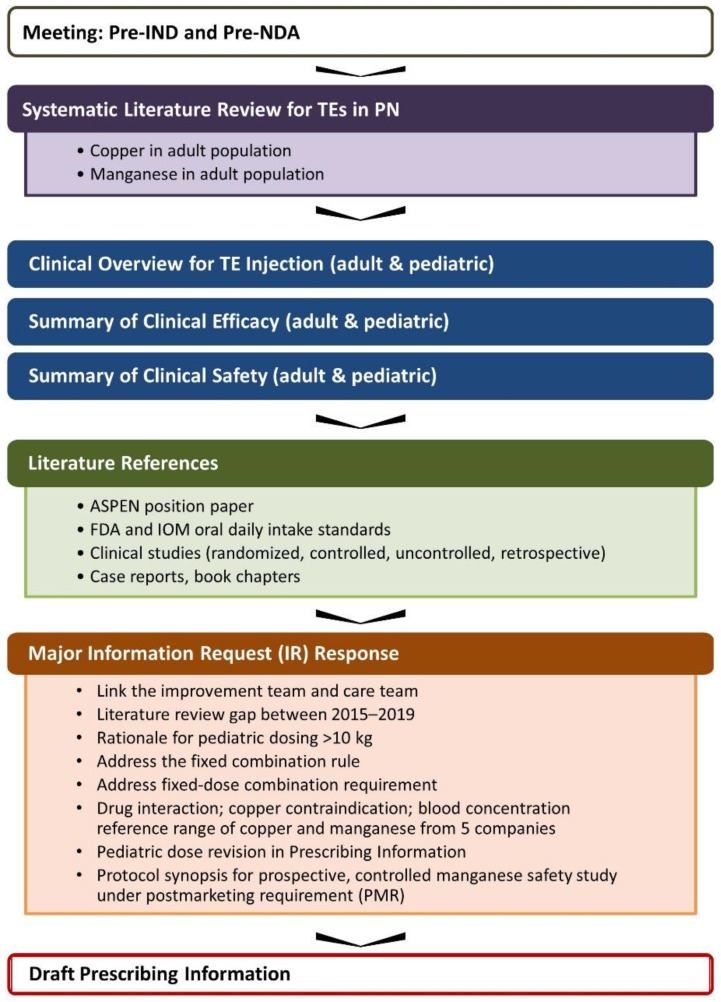
Summary of review materials and analysis included in literature assessments for 505(b)(2) NDA 209376 submission. ASPEN, American Society for Parenteral and Enteral Nutrition; FDA, United States Food and Drug Administration; IND, investigational new drug application; IR, information request; IOM, Institute of Medicine; NDA, new drug application; PMR, postmarketing requirement; PN, parenteral nutrition; TE, trace element.

**Table 1 nutrients-14-01770-t001:** Physiologic properties of TEs that are routinely included in PN [10,11,12].

Trace Element	Physiologic Role	Signs/Symptoms of Deficiency	Signs/Symptoms of Toxicity
Chromium	Component of metalloenzymes and coenzyme in metabolic reactions associated with glucose homeostasis and insulin resistance	Elevated plasma free fatty acids, glucose intolerance (may be refractory to insulin), hyperlipidemia, peripheral neuropathy, and weight loss	Not well documented; potential renal consequences in infants and children
Copper	Enzymatic cofactor involved in connective tissue formation, hematopoiesis and iron metabolism, and CNS function	Hair depigmentation, myocardial disease, neurologic abnormalities, pancytopenia, and skeletal abnormalities	Acute: acute kidney injury, death, diarrhea, hepatic necrosis, and vomitingChronic: cirrhosis, neurological disorders, and renal insufficiency
Manganese	Component of several metalloenzymes and required to activate enzymatic reactions involved in immune function, reproductive health, development of bone and connective tissue, neuronal function, and antioxidant activity	Not well documented	Neuropsychiatric and Parkinson-like symptoms
Selenium	Essential component of selenoproteins involved in anti-inflammatory, antioxidant, and immunological activity, as well as enzymes involved in regulating thyroid hormone metabolism	Cardiac and skeletal muscle myopathy, hair and nail abnormalities, macrocytic anemia, and impaired immune function	Brittle hair and nails, fatigue, GI symptoms, peripheral neuropathy, and skin rash
Zinc	Ubiquitous component of ~120 enzymes with catalytic, structural, and regulatory roles	Alopecia, diarrhea, eye and skin lesions, growth retardation, reduced immune function, and increased susceptibility to oxidative damage	Acute: high oral doses can cause abdominal pain, diarrhea, and vomitingChronic: high oral doses can cause low copper by interfering with copper absorption

CNS, central nervous system; GI, gastrointestinal; PN, parenteral nutrition; TE, trace element.

**Table 2 nutrients-14-01770-t002:** Trace element PN products commercially marketed in the US over time.

Product ^a^	Manufacturer	FDA Approved (NDA Year)	Marketed	Cu mg/mL	Cr mcg/mL	Mn mcg/mL	Se mcg/mL	Zn mg/mL
Peditrace^® b^ [8]	Fresenius Kabi	No	TI	0.02	—	1	2	0.25
Addamel N^® b^ [8]	Fresenius Kabi	No	TI	0.13	1	27	3.2	0.65
4-Trace Elements^® b^ [8]	Hospira	No	TI	0.2	2	16	—	0.8
Multitrace-4 Neonatal [17]	American Regent	No	Yes	0.1	0.8	25	—	1.5
Multitrace-4 Pediatric [17]	American Regent	No	DC	0.1	1	25	—	1
Multitrace-4 [17]	American Regent	No	DC	0.4	4	100	—	1
Multitrace-5 [18]	American Regent	No	DC	0.4	4	100	20	1
Tralement^®^ [23]	American Regent	Yes (2020)	Yes	0.3	—	55	60	3
Multrys^TM^ [24]	American Regent	Yes (2021)	Yes	0.06	—	3	6	1
Chromium Chloride [21]	Pfizer	Yes (1986)	Yes	—	4	—	—	—
Cupric Chloride [20]	Pfizer	Yes (1986)	Yes	0.4	—	—	—	—
Manganese Chloride [22]	Pfizer	Yes (1986)	Yes	—	—	100	—	—
Selenious Acid [15]	American Regent	Yes (2019)	Yes	—	—	—	60	—
Zinc Chloride [19]	Pfizer	Yes (1986)	Yes	—	—	—	—	1
Zinc Sulfate [16]	American Regent	Yes (2019)	Yes	—	—	—	—	1
Zinc Sulfate [16]	American Regent	Yes (2019)	Yes	—	—	—	—	3
Zinc Sulfate [16]	American Regent	Yes (2019)	Yes	—	—	—	—	5

All American Regent products manufactured in Shirley, NY, USA. Cr, chromium; Cu, copper; DC, discontinued; FDA, United States Food and Drug Administration; Mn, manganese; NDA, new drug application; PN, parenteral nutrition; Se, selenium; TI, temporary import; US, United States; Zn, zinc. ^a^ Elemental concentrations are reported, not the salt concentrations. Higher concentrations are available for some of the PN products to use in compounding. ^b^ Temporary import status to the United States granted by the FDA due to shortage of products (May 2013).

**Table 3 nutrients-14-01770-t003:** Recommend daily amounts of TEs routinely included in PN [35,36].

Patients/Recommendations	Chromium	Copper	Manganese	Selenium	Zinc
Infants and children (10–40 kg), mcg/kg/day ASPEN recommendations	0.2 (max 5 mcg/day)	20 (max 500 mcg/day)	1 (max 55 mcg/day)	2 (max 100 mcg/day)	50 (max 5000 mcg/day)
Children and adolescents (>40 kg) ASPEN recommendations	5–15 mcg/day	200–500 mcg/day	40–100 mcg/day	40–60 mcg/day	2–5 mg/day
Adults ASPEN recommendations	≤10 mcg/day	0.3–0.5 mg/day	55 mcg/day	60–100 mcg/day	3–5 mg/day

ASPEN, American Society for Parenteral and Enteral Nutrition; PN, parenteral nutrition; TEs, trace elements.

**Table 4 nutrients-14-01770-t004:** Benefits and potential risks associated with using multi-trace element (TE) products versus multiple individual TE products for parenteral nutrition formulations ^a^.

Advantages or Benefits	Disadvantages or Risks
Safe and effective dosage of all 4 TEs in a single administration for adult and pediatric patients weighing 10 kg and above Simplified preparation and administration process with fewer steps and less resources consumed Reduced risk of dosage calculation errors and potential avoidance of medication errors from multiple dosing calculations Reduced risk of microbial contamination by limiting preparation steps/simpler preparation	Inadequate provision of the recommended dosage of the 4 TEs in pediatric populations ° Some pediatric patients may need additional supplementation of single-entity TE products such as zinc, copper, and/or selenium Inadequate provision of the 4 TEs in certain patient populations or conditions, such as those with excessive GI loss or burn lesions Hepatic accumulation of copper and/or manganese over time with inclusion of copper and/or manganese in patients with liver diseases Manganese neurotoxicity ° Despite significant reduction in the manganese dosage as compared with previous formulations, uncertainty remains about the safety of the recommended manganese dosage over time

GI, gastrointestinal; TE, trace element. ^a^ Adapted from US Food and Drug Administration. NDA multi-disciplinary review and evaluation–NDA 209376, TRALEMENT^®^ (trace elements injection 4*). Available online: https://www.fda.gov/media/142354/download. Accessed on 27 August 2021 [37].

**Table 5 nutrients-14-01770-t005:** Factors that may affect TE needs of patients receiving PN [12,36,38].

TE	Chromium	Copper	Manganese	Selenium	Zinc
Patients/conditions that may require TE dose reduction	Patients with renal insufficiency	Patients with cholestasis	Patients with cholestasis	N/A	N/A
Patients/conditions that may require TE dose increase	Pregnant patients; patients who have extensive short bowel resection	Patients with burns or high GI losses (e.g., nasogastric suctioning, diarrhea, ostomy outputs) after gastric bypass or proximal jejunum resection	N/A	Patients with burns, critical illness, continuous renal replacement therapy, high urine output, fistula output/diarrhea, and multiple drains	Patients with burns, high GI losses, sepsis, or hypercatabolic states; patients who have undergone proximal jejunum resection or have extensive short bowel resection

GI, gastrointestinal; N/A not applicable; PN, parenteral nutrition; TE, trace element.

## Data Availability

Not applicable.

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
