# Peer review of "Advances in Trace Element Supplementation for Parenteral Nutrition"

_nutrients, 2022, doi:10.3390/nu14091770_

Round 1
Reviewer 1 Report
Perks et al present a review on recent advances in trace element (TE) supplementation for parenteral nutrition (PN). After discussing the physiological role of TEs, the authors describe availability of TEs for PN from a US historical perspective, and present current A.S.P.E.N. recommendations for TE PN formulations. The next chapter focuses on Tralement® by American Regent, a TE containing product, which has received FDA approval in 9/2020. Finally, the authors address some practical aspects of clinical TE PN management including product preparation, availability, requirements and monitoring.
Comment:
The work by Perks et al is important and deserves publication. Two possibilities exist:
a) Since the importance of Tralement® in PN is largely presented from an US perspective, the manuscript should be submitted to a North American journal (JPEN, AJCN) including minor changes (e.g., there is no chapter #4!).
b) If the authors want to reach a broader international (European) audience (via publication in Nutrients), major changes are necessary:
- Those parts of the manuscript should be deleted which describe aspects specific for the US (Historical Perspective, Rationale and FDA Regulatory Review and Approval of Tralement®, Studies on Product Sterility, Compatibility, and Stability in PN Admixtures)
- Contents, dosing and handling of Tralement® must be compared in detail with its European competitor, Addamel N® by Fresenius Kabi. This is particularly important because recommendations for daily use and dosing differ significantly.
- The authors must discuss in detail the use of Tralement® and Addamel N® in light of the active ESPEN guidelines on the subject for adults and children (1, 2).
- The TEs in question (zinc, copper, manganese and selenium) should be addressed separately in greater detail devoting specific attention to individual complementation, repletion, supplementation, pharmacological dosing and monitoring in well-described clinical entities (short- and long-term TPN, malnutrition, injury and infection, critical illness) and during defined modalities of medical nutrition therapy (total PN vs. partial PN combined with EN vs. fasting).
- Finally, A.S.P.E.N. and ESPEN key recommendations should be compared to each other addressing specific strengths and weaknesses.
References
- Berger MM et al; ESPEN micronutrient guideline. Clin Nutr 2022 https://doi.org/10.1016/j.clnu.2022.02.015
https://www.sciencedirect.com/science/article/pii/S0261561422000668
- Domellöf M, Szitanyi P, Simchowitz V, Franz A, Mimouni F; ESPGHAN/ESPEN/ESPR/CSPEN working group on pediatric parenteral nutrition. ESPGHAN/ESPEN/ESPR/CSPEN guidelines on pediatric parenteral nutrition: Iron and trace minerals. Clin Nutr. 2018 Dec;37(6 Pt B):2354-2359.
Reviewer 2 Report
Perks et al. review provides readers with information on the evolution of supplemental PN in the United States as well as information on the use of parenteral nutrition in patients. However, multiple aspects were left out of the manuscript.
- No information was found about clinical disorders that cause people to absorb these nutrients insufficiently.
- Does the gut microbiota play a role in poor nutrient absorption or do they aid absorption?
- Furthermore, the paper failed to disclose ongoing clinical trials of supplemented PN.
- Furthermore, the article only mentioned a few hazardous effects of these substances when given in excess. If not given in sufficient amounts, they can be fatal. It is preferable to concentrate on those aspects.
- Some of the names of the companies are mentioned repeatedly in the article.
- Are there any new regulations that regulatory authorities are enforcing for the manufacture of such supplements?
Reviewer 3 Report
A very interesting publication. It is a pity that the problem of supplementation with trace elements in newborns (especially premature babies) and infants was omitted. It is a very little-understood topic that requires further research and study.
Round 2
Reviewer 1 Report
It is unfortunate that the authors did not address any of my comments; since the authors insist to present a US experience in relation to the development of compounded PN admixtures in coordination with evolving ASPEN and FDA requirements, it is more appropriate to present this information in a North American journal
Reviewer 2 Report
The manuscript can be accepted in the current form.